# Tuberculous Fibrosis Enhances Tumorigenic Potential via the NOX4–Autophagy Axis

**DOI:** 10.3390/cancers13040687

**Published:** 2021-02-08

**Authors:** Seong Ji Woo, Youngmi Kim, Harry Jung, Jae Jun Lee, Ji Young Hong

**Affiliations:** 1Institute of New Frontier Research, Hallym University College of Medicine, Chuncheon, Gangwon-do 24253, Korea; seong-jikr@nate.com (S.J.W.); kym8389@hanmail.net (Y.K.); harry880219@gmail.com (H.J.); iloveu59@hallym.or.kr (J.J.L.); 2Lung Research Institute, Hallym University College of Medicine, Chuncheon, Gangwon-do 24253, Korea; 3Division of Pulmonary, Allergy and Critical Care Medicine, Department of Internal Medicine, Chuncheon Sacred Heart Hospital, Hallym University Medical Center, Chuncheon-si, Gangwon-do 24253, Korea

**Keywords:** autophagy, lung cancer, NOX4, tuberculous fibrosis

## Abstract

**Simple Summary:**

Although previous studies have reported coexistence of pulmonary TB and carcinoma, the underlying mechanism of tuberculous fibrosis-induced tumorigenicity remains to be investigated. We previously reported that NOX4 signaling mediates tuberculous pleural fibrosis by activating ERK–ROS–EMT pathways. We were interested in the role of NOX4 in the tumor microenvironment changed by tuberculosis fibrosis. Our results showed that lung cancer cells enhanced the NOX4 expression and invasive potential after exposure to the conditioned medium of heat-killed *Mycobacterium tuberculosis* stimulated mesothelial cells or tuberculous pleural effusion. NOX4–autophagy signaling axis contributes to the interaction between tuberculosis fibrosis and lung cancer. Silencing of NOX4 signaling in tuberculous fibrosis reduced the metastatic potential by enhancing autophagy in both in vivoand in vitro studies. This result suggests that NOX4-P62 might serve as a therapeutic target for tuberculous fibrosis-associated lung cancer.

**Abstract:**

While a higher incidence of lung cancer in subjects with previous tuberculous infection has been reported in epidemiologic data, the mechanism by which previous tuberculosis affects lung cancer remains unclear. We investigated the role of NOX4 in tuberculous pleurisy-assisted tumorigenicity both in vitro and in vivo.Heat-killed *Mycobacterium tuberculosis*-stimulated mesothelial cells augmented the migrationand invasive potential of lung cancer cells in a NOX4-dependent manner. Mice with *Mycobacterium bovis* bacillus Calmette–Guérin (BCG) pleural infection exhibited increased expression of NOX4 and enhanced malignant potential of lung cancer compared to mice with intrathoracic injection of phosphate-buffered saline. The BCG+ KLN205 (KLN205 cancer cell injection after BCG treatment) NOX4 KO mice group showed reduced tuberculous fibrosis-promoted metastatic potential of lung cancer, increased autophagy, and decreased expression of TGF-β, IL-6, and TNF-α compared to the BCG+KLN205 WT mice group. Finally, NOX4 silencing mitigated the malignant potential of A549 cells that was enhanced by tuberculous pleural effusion and restored autophagy signaling. Our results suggest that the NOX4–autophagy axis regulated by tuberculous fibrosis could result in enhanced tumorigenic potential and that NOX4-P62 might serve as a target for tuberculous fibrosis-induced lung cancer.

## 1. Introduction

Tuberculosis (TB) is a leading infectious cause of death [1]. Chronic host–pathogen interaction in TB causes broad remodeling of lung tissue [2]. In tuberculous pleurisy, which manifests in up 30% of patients with TB, pleural fibrosis commonly occurs afterpleural inflammation [3,4]. The progression of pleural fibrosis leads to severe clinical manifestations such as lung entrapment and respiratory failure. Pleural mesothelial cells (PMCs) play a critical role in pleural fibrosis by upregulating vasoactive substances and inducing proliferation of the extracellular matrix in response to pleural inflammation [4].

We previously found that NOX4 signaling mediates tuberculous pleural fibrosis via ERK-reactive oxygen species (ROS) signaling [5]. Furthermore, some studies have reported that NOX4 signaling reinforces the growth and invasion of lung cancer via positive feedback regulation of PI3K/Akt signaling [6,7]. This led us to hypothesize that NOX4 may be a link between tuberculous fibrosis and lung cancer.

Studies have examined the association between carcinogenesis and genetic damage caused by infection-induced inflammation and fibrosis [8,9,10]. Several pathologies such as *Helicobacter pylori* or inflammatory bowel diseases support the idea that chronic inflammation may enhance the risk of cancer [9,10]. Many population-based studies reported coexistence of pulmonary TB and carcinoma [11,12,13,14]. One experimental study showed that chronic TB lung lesions result in an environment propitious to carcinogenesis [2]. Production of ROS and prostaglandin, increased rates of cell division, and increased oxidative DNA damage have been suggested as possible mechanisms of scar carcinoma formation [15]. However, the mechanism underlying extensive fibrosis-associated TB infection and lung cancer remains to be evaluated.

In this study, we examined the effect of heat-killed *Mycobacterium tuberculosis* (HKMT)-infected PMCs on the induction of the epithelial–mesenchymal transition (EMT), migration, and invasion of lung cancer cells using an in vitro co-culture method. A *Mycobacterium bovis* bacillus Calmette–Guérin (BCG)-induced pleurisy mouse model was used to explore the role of NOX4 in tuberculous pleurisy-assisted metastatic potential of lung cancer. We assessed the effects of downregulating NOX4 signaling on tuberculous effusion-treated A549 cells in regulation of autophagy signaling and invasiveness of lung cancer.

## 2. Results

### 2.1. Downregulation of NOX4 Signaling Reduces HKMT-Induced Collagen and POLDIP2Synthesis by Activating Autophagy Signaling

A previous study showed that NOX4 signaling mediates the EMT process and regulates autophagy signaling in tuberculous fibrosis [5]. We confirmed that 10 ng/mL HKMT treatment for 1 h induced NOX4 production in mesothelial cells (Appendix A).

POLDIP2 is a NOX4 regulatory protein that has a role in ROS production and cytoskeletal remodeling [16]. Collagen is an extracellular protein that is used to evaluate lung fibrosis [5]. In vitro, HKMT treatment enhanced the expression of NOX4 along with POLDIP2 and collagen (Figure 1). The concentrations of TGF-β, IL-6, and TNF-α in conditioned medium of Met5A cells following HKMT treatment were higher than those without HKMT treatment. The expression of LC3II/LC3Iwas lower and the expression of p62 was higher in Met5A cells with HKMT treatment compared to untreated Met5A cells.

Downregulation of NOX4 inhibited the expression of POLDIP2 and collagen. Lower levels of TGF-β, IL-6, and TNF-αwere detected in the SiNOX4+HKMT group compared with the SiCon+ HKMT group. Western blotting and fluorescent microscopy showed that SiNOX4 treatment prevented the downregulation of LC3II/LC3I and upregulation of p62. Consistent results were observed in the BCG-induced pleurisy mouse model. After BCG treatment, WT mice showed lower expression of LC3II and ATG12-ATG5 in lung tissue than untreated WT mice. The expression of LC3II/LC3I and ATG12-ATG5 increased in NOX4 KO mice with BCG treatment compared to WT mice with BCG treatment. Immunohistochemical staining of lung tissue showed that downregulation of NOX4 decreased the BCG-induced expression of p62. These results suggest that NOX4 signaling regulates autophagy signalingin tuberculous fibrosis.

### 2.2. HKMT-Induced PMCs Enhance Migration and Invasion of Lung Cancer Cells via NOX4 Signaling

Reports have suggested that NOX4 may be a potential therapeutic target against both non-small cell lung cancer and tuberculous fibrosis [5,6,7]. Thus, we evaluated whether tuberculous fibrosis promotes the malignant potential of lung cancer and whether NOX4 regulates the cellular interaction between PMCs and lung cancer cells. Wound healing assays showed that conditioned medium of mouse PMCs after HKMT treatment increased lung cancer cell migration, whereas suppression of NOX4 significantly inhibited lung cancer cell migration. The Transwell assay revealed similar results. Lung cancer cells were seeded on Matrigel for 48 h and conditioned mouse PMC medium with or without HKMT treatment was used as a chemoattractant in the lower chamber. Lung cancer cells were stained with H&E staining solution to visualize cells that had penetrated the Matrigel and migrated to the lower chamber. The invasiveness of lung cancer cells treated with conditioned medium of HKMT-treated PMCs was significantly higher than that of untreated PMCs (Figure 2A). Furthermore, downregulation of NOX4 using PMCs from NOX4 KO mice resulted in reduced invasiveness compared with mesothelial cells treated only with HKMT in the Transwell assay (*p* < 0.001).

Wound healing assays and the Transwell assay using conditioned medium of human Met5A cells showed the consistent results (Figure 2B). Downregulation of NOX4 was performed by siNOX4 treatment.

### 2.3. NOX4 Is Required for the Increased Metastatic Potential of Cancer Cells by Tuberculous Fibrosis

The role of NOX4 on the metastatic potential of lung cancer cells was investigated. Atday 29 after BCG injection, NOX4 WT and KO mice were injected with 2 × 10^5^ lung cancer cells and sacrificed on day 43. BCG increased the metastatic potential of lung cancer cells in a NOX4-dependent manner (Figure 3A). Macroscopic examination using Bouin’s fixative solution showed BCG treatment increased the number of lung metastatic nodules on the surface compared to PBS treatment. H&E staining showed the consistent results. NOX4KO/PBS/KLN205 mice (NOX4 KO mice with KLN205 cancer cell injection after BCG treatment) showed a reduced number ofclusters of squamous lung cancer cells and decreased expression of Ki67 compared to NOX4WT/BCG/KLN205 mice (NOX4WT mice with KLN205 cancer cell injection after BCG treatment) (Figure 3B). Appendix A demonstrates that fibrosis is markedly increased in mice with BCG treatment compared to PBS treatment (WT/PBS/KLN205 vs WT/BCG/KLN205) and lung fibrosis after BCG injection is reduced in NOX4 KO mice compared to WT mice (NOX4 KO/BCG/KLN205 vs WT/BCG/KLN205). These findings are consistent with a previous study [5].

NOX4–autophagy signaling is involved in tuberculous fibrosis-induced cancer progression. In this process, NOX4 signaling controls Snail, the inducer of EMT process. Compared to NOX4WT/PBS/KLN205 mice, NOX4WT/BCG/KLN205 mice showed increased expression of p62 and Snail but decreased expression of ATG12-5 and LC3II/LC3I. In NOX4KO/BCG/KLN205 mice, expression of p62 and Snail was decreased while that of ATG12-5 and LC3II/LC3Iwas increased compared with NOX4WT/BCG/KLN205 mice (Figure 3C). We also assessed p62 levels by immunofluorescence confocal microscopy and found increased levels of p62 in NOX4WT/BCG/KLN205 mice than NOX4WT/PBS/KLN205 and NOX4KO/BCG/KLN205 mice, consistent with the Western blot results (Figure 3D).

TNF-α, IL-6, and TGF-β are inflammatory cytokines that participate in both the initiation and progression of cancer [17]. Concentrations of the cytokines TGF-β, IL-6, and TNF-α in lung tissue lysates and serum were higher in NOX4WT/BCG/KLN205 mice than NOX4WT/PBS/KLN205 mice. Decreased cytokine expression was detected in lung tissue lysates and the serum of NOX4KO/BCG/KLN205 mice compared withNOX4WT/BCG/KLN205 mice (Figure 3E). Taken together, NOX4 contributes to the increased metastatic potential of cancer cells by tuberculous fibrosis through autophagy signaling.

### 2.4. Autophagy Is Attenuated in Lung Cancer Cells after Tuberculous Effusion Treatment

Pleural effusion of patients with tuberculous pleurisy was used to examine the association between tuberculous fibrosis and lung cancer. Electron microscopy, the gold standard for the identification of autophagosomes, confirmed the Western blotting and immunofluorescence microscopy results described above (Figure 4). The SiCon-treated (control) and SiNOX4-treated groups exhibited numerous autophagosomes (black arrows). The number of autophagosomes was significantly decreased in the SiCon + TPE (tuberculous pleural effusion) group than the SiCon group (*p* < 0.01). Interestingly, the SiNOX4 + TPE group demonstrated an increased number of autophagosomes than the SiCon + TPE group (*p* < 0.05).

### 2.5. TPE Enhances Migration, Invasion, and the EMT in Lung Cancer Cells in a NOX4-Dependent Manner

We tested whether TPE promotes invasion and migration of lung cancer (Figure 5). TPE induced the migration of A549 cells; suppression of NOX4 reduced the TPE-induced migration of A549 cells. Consistent with the wound healing assay, the Transwell assay revealed that the invasiveness of A549 cells co-cultured with TPE was significantly increased compared to those without TPE (*p* < 0.001). Inhibition of NOX4 signaling in A549 cells by RNA interference abrogated the increase in invasive ability.

It is well known that both EMT and autophagy are the main biological processes in cancer initiation, progression, and metastasis [18].In the EMT process, Snail increases as a main regulator to promote the transcription of genes expressed in mesenchymal cells, while the epithelial marker Zonaoccludin-1 (ZO-1)is suppressed [19]. Corroborating the Western blotting data, TPE-treated A549 cells increased the expression of p62 and Snail but reduced the expression of LC3II/LC3I, ATG12-ATG5, and ZO-1. However, with the inhibition of NOX4, this effect was reversed with the upregulated expression of LC3II/LC3I and ATG12-ATG5 and downregulated expression of p62 and Snail. This result suggests the involvement of NOX4–autophagy signaling in the TPE-induced EMT in A549 cells. Contrary to TPE, transudate resulted in decreased expression of NOX4 and increased autophagy signaling, leading to lower metastatic potential of A549 cells (Appendix A).

## 3. Discussion

Chronic infection and consequent inflammation activates the EMT, which leads to cancer metastasis and fibrinogenesis [8]. *H*. *pylori* CagA destabilizes E-cadherin/βcatenin and promotes the EMT [20]. Diffusely adherent *Escherichia coli* induces the EMTby stimulating IL-8 and vascular endothelial growth factor in enterocytes [21].

Several epidemiological studies have shown that the coexistence rate of pulmonary TB and lung cancer is high [11,12,13,14]. Dacosta et al. found that TB-associated lesions, either active or inactive, were detected in 29 of 96 necropsy cases (30.2%) with bronchogenic carcinomas, compared with 7% of the general population [11]. Female patients with pulmonary TB in the Nagoya TB registry were at a very high risk of death from lung cancer [12].

Nalbandian et al. showed that TB induced DNA damage and produced the epidermal growth factor epiregulin responsible for tumorigenesis [2]. *M. tuberculosis*-infected THP-1 cells have been shown to promote the EMT process in A549 cells via the JNK/MAPK pathway [22]. Holla et al. showed that activation of SHH signaling by BCG downregulated p53 and inhibited apoptosis in A549 cells. However, the casual link and detailed mechanisms between TB-associated lung lesions and lung cancer are still poorly studied [23].

In this study, we provide experimental evidence that NOX4signaling activated by tuberculous fibrosis creates a microenvironment conducive to tumor progression. NADPH oxidase enzymes play a critical role in generating ROS in normal physiological processes and various diseases [24]. Especially, NOX4 contributes to the pathogenesis of various lung diseases such as asthma, idiopathic pulmonary fibrosis, and *Pseudomonas aeruginosa* lung infection [25,26,27]. We found that HKMT enhances the expression of POLDIP2 and collagen in PMCs in a NOX4-dependent manner.POLDIP2 is involved in cytoskeletal integrity in smooth muscle cells and kidney myofibroblasts as a positive regulator of NOX4 [16,28]. As previously described by our group [5], inhibiting NOX4 signaling induces cytoprotective autophagy and resolution of the EMT in tuberculous fibrosis.

The conditioned medium of HKMT-stimulated PMCs increased the migration and invasion of lung cancer cells in vitro (Figure 2). Levels of TGF-β, IL-6, and TNF-α were increased in the conditioned medium of HKMT-stimulated PMCs and were reduced by NOX4 inhibition. TGF-β, IL-6, and TNF-α play a role in tumor formation in conditions of chronic inflammation [17]. These cytokines contribute to mutation, tumor immune evasion, angiogenesis, and metastasis [17]. TGF-β and IL-6 have been recognized as key drivers in several carcinomas [29,30,31]. TNF-α induces the EMT in several carcinomas via transcription-factor-dependent mechanisms [32,33], supporting the interaction between TB fibrosis and lung cancer in the present study.

Likewise, in the in vivo BCG-induced pleurisy model, BCG treatment enhanced the metastatic potential of lung cancer. TGF-β, IL-6, and TNF-α were increased in the sera and lung lysates of mice with BCG injection and subsequent lung cancer cell injection (Figure 3). We have thus demonstrated NOX4as a promising therapeutic target. Downregulating NOX4 signaling reduced the infiltration of lung cancer cells and the expression of proinflammatory cytokines with restoration of autophagosomes in lung tissues. In humans, TPE has been shown to exhibit higher NOX4 levels compared to transudate and to increase migration and invasiveness of lung cancer cells. Consistent with this, NOX4 silencing increased autophagic flux and decreased the migration and invasiveness of lung cancer cells. To the best of our knowledge, this is the first study demonstrating that tuberculous fibrosis promoted the EMT and invasiveness of lung cancer via NOX4/autophagy signaling (Figure 6).

The role of NOX4 in regulating autophagy appears to differ depending on cell type [34]. Some studies have reported that NOX4 induces autophagy [35,36]. NOX4-induced autophagy was reported to trigger anticancer drug resistance in head and neck cancer [35]. The anti-cancer activity of coumarin in breast cancer has been linked to protective autophagy through NOX4-dependent ROS production [36]. In contrast, our data indicate that silencing of NOX4 signaling in tuberculous fibrosis reduced the metastatic potential by enhancing autophagy, similar to the study of Liu et al., which showed that NOX4 inhibits autophagy, resulting in enhanced invasive properties of renal cell carcinoma cells [37].

## 4. Materials and Methods

### 4.1. Cell Lines and Animals

The human PMC lineMet5A, human adenocarcinoma cell line A549, and mouse musculus lung squamous cell line KLN205 were purchased from the American Type Culture Collection (ATCC, Manassas, VA, USA) and cultured under the manufacturer’s instructions.

Wild-type (WT) C57BL/6 mice were purchased from DooYeol Biotech (Seoul, Korea) and NOX4 knockout (KO) C57BL/6 mice were obtained from Prof. Park (Yonsei University, Seoul, Korea). Mice were bred in the vivarium of HallymUniversity (Chuncheon, Korea). All animal experiments were approved by the Institutional Animal Care and Use Committee of Hallym University (No: Hallym 2017-47).

### 4.2. Mouse Pleural Mesothelial Cells

Isolation of mouse PMCs was conducted as previously described [5]. Primary mouse PMCs were freshly isolated from the lungs and heart of 3–4-week-old mice. Cells were cultured for 7 days in Dulbecco’s modified Eagle’s medium (Gibco, Waltham, MA, USA) supplemented with 1% penicillin/streptomycin and 15% fetal bovine serum. Cells were washed with phosphate-buffered saline (PBS) and the medium was replaced every 2 days. WT and NOX4 KO PMCs were used at passages 2–3 when a homogeneous population of cobblestone PMCs was identified.

### 4.3. Heat-Killed M. tuberculosis Treatment, Cell Transfection, and Immunofluorescence of LC3

HKMT was purchased from InvivoGen (San Diego, CA, USA). A small interfering RNA (siRNA; BioneerInc, Daejeon, Korea) against NOX4 based on the target region of the NOX4 gene (sense:5′-UCAGACAAAUGUAGACAC-3′ and antisense: 5′-AGUGUCUACAUUUGUCUG-3′; siNOX4) and scrambled siRNA (sense:5′-GGTCAAGACACTATTAACA-3′ and antisense:5′-GGATTCCTAGTGTATTTCA-3′; SiCon) were used in experiments. The mesothelial cells were treated with 10 ng/mL HKMT for the 1 h after transfection with siCon or SiNOX4.

To assess the effect of NOX4 on HKMT-induced autophagosome formation in Met5A, LC3 puncta was measured using immunofluorescence staining. In brief, Met5Acells were fixed with 4% paraformaldehyde solution and then blocked with 2.5% normal goat serum blocking solution (Vector Laboratories, Burlingame, CA, USA) for 1 h. The samples were treated with the antibody anti-LC3 (Cell signaling Technology, USA) overnight at 4 °C, followed by treatment with secondary antibody Alex Fluor 488 donkey anti-rabbit IgG (Gibco, Grand Island, NY, USA) for 1 h at 37℃.Finally, cells were rinsed with PBS and examined under a fluorescent microscope (Olympus FV500; Olympus, Tokyo, Japan). The number of LC3 puncta was counted in four separate fields.

### 4.4. Human Pleural Effusion Collection

The retrospective study was conducted at Chuncheon Sacred Heart Hospital in Korea from January 2014 to June 2019. Thestudy protocol was approved by the Institutional Review Board of Chuncheon Sacred Heart Hospital (IRB no.2012-27). All study participants provided written informed consent for inclusion in this study. Tuberculous pleural effusion (TPE) was defined by two criteria: positive culture result from pleural fluid sample or pleural biopsy with mycobacterial histological features. Transudate pleural effusion was defined by Light’s criteria [38]. A group of samples (*n* = 5) were collected and used for in vitro treatment of A549 cells. Appendix A shows the characteristics of pleural effusion samples. Pleural effusion was prepared as previously described [39].

### 4.5. Effect of BCG-Induced Pleural Fibrosis on Metastatic Potential

For induction of BCG-induced pleural fibrosis, WT and KO NOX4 mice (6–8 weeks old) were injected with 10^6^ colony forming units (CFUs) ofBCG Pasteur in 100 µL of PBS in the intrapleural cavity. After 4weeks, 2 × 10^5^ mouse musculus lung squamous cells (KLN205; ATCC) were intravenously injected. Two weeks after KLN205 injection, mice were anesthetized with 4% isoflurane (Piramal Critical Care, Bethlehem, PA, USA) before lung tissue and sera were obtained. All animal experiments were conducted in a laboratory affiliated with Institute of New frontier Research.

### 4.6. Scratch-Wound Assay and Cell Invasion Assay

For the scratch-wound assay, lung cancer cells were seeded to create a confluent monolayer in 6-well plates. After making a scratch on the cell layer with a micropipette tip, we analyzed photographs taken at 0 and 48 h quantitatively by measuring the distances between one side of scratch and the other. To evaluate the effect of tuberculous fibrosis on the migration potential of lung cancer, the conditioned medium of mesothelial cells after HKMT treatment was added to lung cancer cells cultured in serum-free medium at a 1:1 ratio.

A cell invasion assay was performed using a 24-well Transwell chamber with a polycarbonate membrane of pore size of 8 μm (Corning, Inc., Corning, NY, USA). RPMI-1640 containing lung cancer cells were plated in the upper Transwell chamber. The conditioned medium of mesothelial cells or pleural effusion was added to the lower Transwell chamber. After 48 h, the lung cancer cells that had penetrated the Matrigel barrier to the lower surface were stained with crystal violet solution or hematoxylin and eosin (H&E) staining solution and counted.

### 4.7. Real-Time Reverse Transcription PCR, Western Blotting, and Enzyme-Linked Immunosorbent Assay

mRNA expression was analyzed by real-time reverse transcription (RT) PCR in accordance with the previous study [37]. The relative expression was calculated as ddCt and data were normalized to the expression of β-actin mRNA for each sample. Appendix A shows the primer sequences for each gene.

Standard Western blotting techniques were used.The following antibodies were used:β-actin(Cell Signaling Technology, Danvers, MA, USA), LC3B(Cell Signaling Technology, Danvers, MA, USA), ATG12(Cell Signaling Technology, Danvers, MA, USA), Snail(Cell Signaling Technology, Danvers, MA, USA), E-cadherin (Cell Signaling Technology, Danvers, MA, USA), NOX4(Santa Cruz Biotechnology, Dallas, TX, USA), p62(Santa Cruz Biotechnology, Dallas, TX, USA), Poldip2(Abcam, Cambridge, UK), and ZO-1 (Abcam, Cambridge, UK). The presence of transforming growth factor (TGF)-β, interleukin (IL)-6, and tumor necrosis factor (TNF)-α in mouse serum and cell conditional mediumwas quantified using enzyme-linked immunosorbent assays (ELISAs; Cusabio, Wuhan, China).

### 4.8. Lung Tissue Staining

Mouse lung tissues were fixed in 10% formaldehyde in PBS and stained with Bouin’s solution (Sigma-Aldrich, St. Louis, MO, USA). After 24 h, the lungs were washed in water and the extent of metastasis was quantified. Paraffin-embedded tissue blocks were sectioned at 4 μm thickness and H&E staining was performed. To evaluate pulmonary fibrosis, samples were stained with Masson’s trichrome (IMEB Inc., Chicago, IL, USA). For immunohistochemical staining, the sections were dehydrated, deparaffinized in xylene, and rehydrated with 100%, 95%, 75%, and 50% ethanol. After blocking with 5% normal goat serum (Vector Laboratories, Burlingame, CA, USA) at room temperature for 1 h, the slides were incubated with primary anti-Ki67 (1:50; Abcam) at 4 °C overnight followed by incubation with goat anti-rabbit secondary antibody (Gibco, Grand Island, NY, USA) for 1 h at room temperature.

For immunofluorescence staining, the sections were incubated with antibody againstp62(1:50; Santa Cruz Biotechnology, Santa Cruz, CA, USA). After washing three times with Tris-buffered saline(TBS),the slides were incubated with Alexa Fluor 594-conjugated donkey anti-mouse IgG1(Gibco, Grand Island, NY, USA)for 1 h in dark. Finally, the slides were washed with TBST (Tris-buffered saline, 0.1% Tween 20) and examined under a fluorescent microscope (Olympus FV500; Olympus, Tokyo, Japan).4′,6-Diamidino-2-phenylindole dihydrochloride(DAPI; Sigma-Aldrich) was used as a counterstain.

### 4.9. Transmission Electron Microscopy

A549 cells treated with TPE or untreated were prepared and examined in afield-emission transmission electron microscope (FE-TEM, JEM 2100F; Jeol, Japan). Modified Karnovsky’s fixative was used to identify autophagosomes [40]. Images were taken and analyzed as previously described [41].

### 4.10. Statistical Analyses

All experiments were repeated three times independently. GraphPad Prism software was used for statistical analyses. Data are reported as the means ± standard error of the mean (SEM). Unpaired Student’s *t*-test with differences between means was used. A value of *p* < 0.05 was considered significant for all statistical comparisons.

## 5. Conclusions

In conclusion, we present evidence that the NOX4-autophagy signaling axis plays a role in the remodeling of the tumor environment that resulted from the interaction of cancer cells with TB-educated PMCs. These findings represent a challenge for interventions in both conditions and are expected to provide new therapeutic targets for lung cancer. Macrophage activationin the TB-associated microenvironment and subsequent production of ROS and inflammatory cytokines may be correlated with the increasedpremalignant potential of cancer cells. Further study is required to figure out the detailed mechanisms of PMCs, macrophages, and cancer cells in association with the NOX4–autophagy signaling axis.

## Figures and Tables

**Figure 1 cancers-13-00687-f001:**
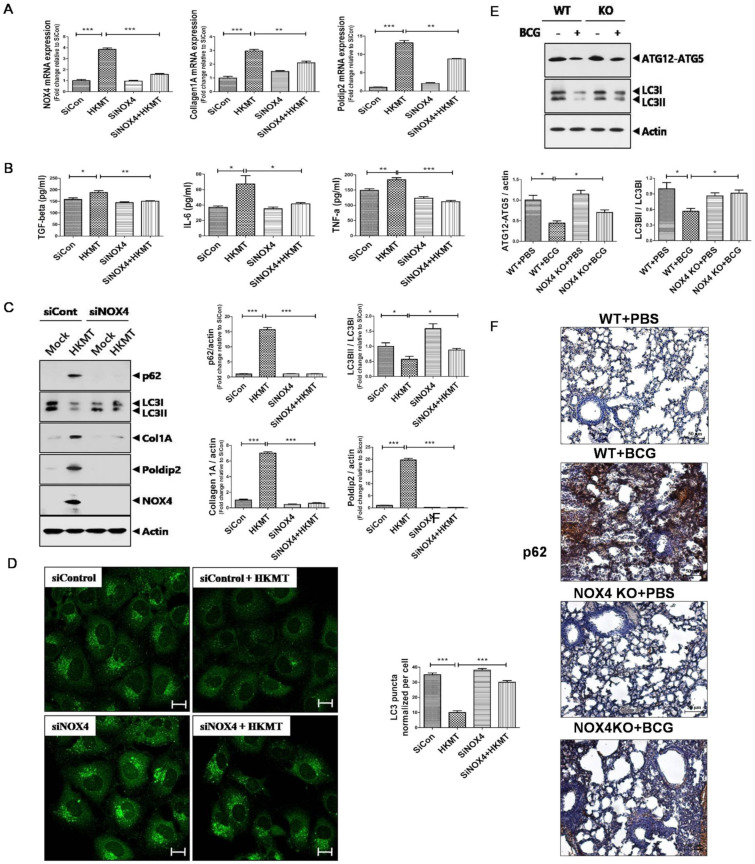
NOX4 mediates HKMT (heat-killed *Mycobacterium tuberculosis*) -induced collagen synthesis via POLDIP2 and autophagy signaling: Human Met5A cells were transfected with siRNA targeting NOX4 (SiNOX4). After 1 day, Met5A cells were treated with HKMT (10 ng/mL) for 1 h or untreated (**A**–**D**). NOX4 mediates autophagy signaling in BCG (bacillus Calmette-Guérin)-induced lung fibrosis (**E**,**F**). (**A**) RT-PCR analysis ofNOX4, POLDIP2, and COL1A mRNA levels. (**B**) Cytokine levels (TGF-β, IL-6, and TNF-a) were assessed by ELISA in conditioned media of Met5A cells following transfection with siCon or siRNA targeting NOX4 (SiNOX4). (**C**) Western blotting of p62, LC3II/LC3I, collagen, POLDIP2, and NOX4. (**D**) LC3-puncta formation with quantification in Met5A cells with different treatments. Images were obtained by confocal microscopy (80×). Scale bar: 50 μm. (**E**) Western blotting of ATG12-ATG5 and LC3B. (**F**) Immunostaining of p62 in mouse lungs (200×). Scale bar: 50 μm. Results are presented as the mean ± SEM of at least three independent experiments. *** *p* < 0.001, ** *p* < 0.01, * *p* < 0.05.

**Figure 2 cancers-13-00687-f002:**
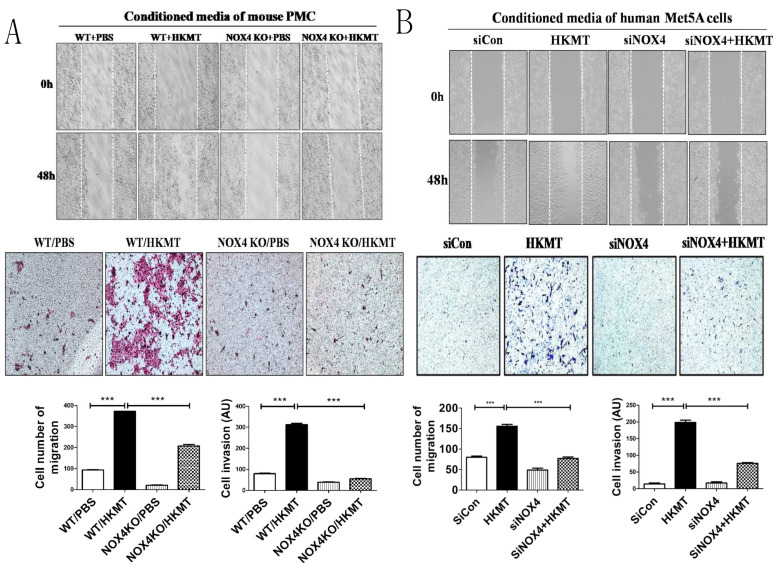
NOX4 regulates tuberculosis-induced lung cancer cell migration and invasion: Scratch-wound healing assay and Transwell assay showed thatNOX4 downregulation significantly prevented the migration and invasion of lung cancer cells. In the Transwell assay, lung cancer cells were platedin the upper chamber containing Matrigel, and the conditioned medium of PMCs (pleural mesothelial cells) with or without HKMT treatment as the chemoattractant in the lower chamber was used for 48 h.(**A**)Mouse model: mouse PMCs and KLN205 lung cancer cells were used. (**B**) Human model: human PMC Met5A cells and A548 lung cancer cells were used. Data are presented as the mean ± SEM of at least three independent experiments. *** *p* < 0.001.

**Figure 3 cancers-13-00687-f003:**
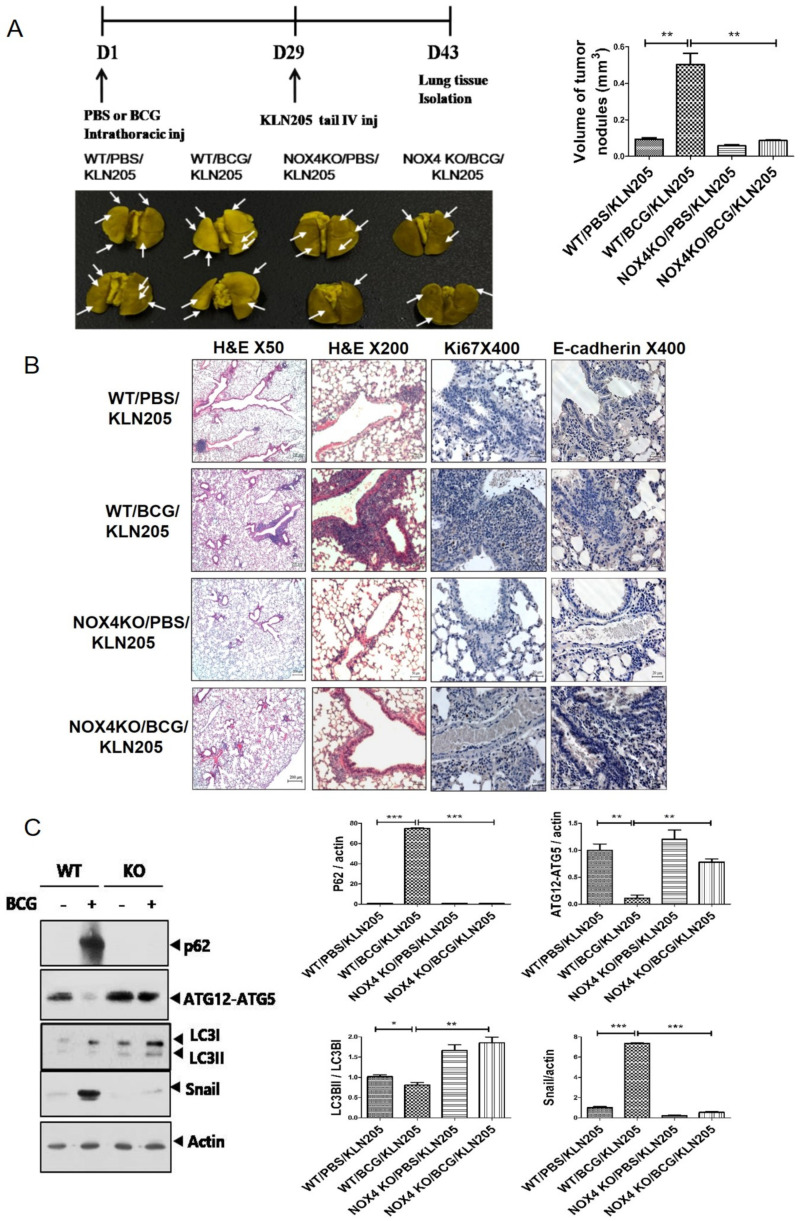
NOX4 mediates BCG pleurisy-promoted increased metastatic potential of lung cancer cells: (**A**) C57BL/6 NOX4 WT and KO mice were administered an intrapleural injection of BCG (1 × 10^6^ CFUs BCG Pasteur in 100 μL of saline). Each mouse also received an intravenous injection of KLN205 mouse lung cancer cells (2 × 10^5^) at day 29. At day 43, lung tissues were harvested. Representative micrographs of Bouin staining of metastatic nodules are presented. (**B**) Histological images of lungs. H&E staining (50×, 200×) and Ki67 immunohistochemistry (400×). Ki-67-positive cells show nuclear brown staining. (**C**) Lung cancer tissue lysates from each mouse were used for Western blotting of P62, ATG5-ATG12, LC3II/LC3I and Snail. (**D**) Activation of p62 deposition is regulated by NOX4 in a BCG pleurisy lung cancer model. p62 was subjected to immunofluorescence staining. Immunofluorescence staining of p62 (red); DAPI staining (blue); Purple represents colocalization of DAPI staining and immunofluorescence of p62. (**E**) RT-PCR analysis of TGF-β, IL-6, and TNF-α mRNA levels in mouse lung tissue lysates. ELISA analysis of TGF-β, IL-6, and TNF-α levels in mouse sera. Eight mice were used in each group (*n*= 8 per group). *** *p* < 0.001, ** *p* < 0.01, * *p* < 0.05.

**Figure 4 cancers-13-00687-f004:**
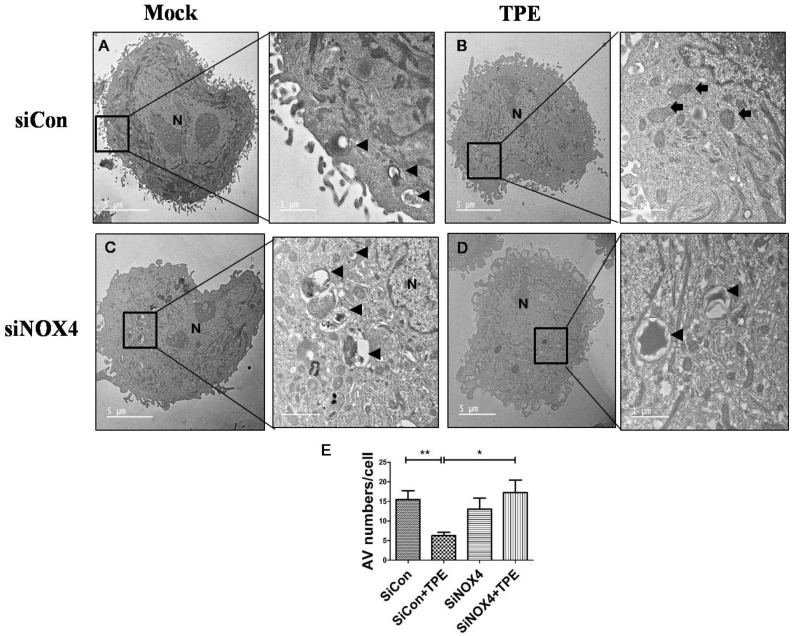
Loss of NOX4 increases autophagic vesicle abundance in A549 cells with TPE (tuberculous pleural effusion):A549 cells were transfected with siCon or siRNA targeting NOX4 (SiNOX4) for 1 day followed by treatment with or without TPE for 48 h. (**A**) siCon (negative control) group; (**B**) siCon+ TPE group; (**C**) siNOX4 group; (**D**) siNOX4+ TPE group. Abundant autophagic vesicles (AVs) were observed in A549 cells but few AVs were observed by transmission electron microscopy following treatment with TPE. Black arrowheads point to AVs including digested material. Black arrows indicate mitochondria. (**E**) The number of autophagic vesicles (AV) in each cell was counted with 5 cells in each sample, respectively. Data indicate the results of three experiments. ** *p* < 0.01, * *p* < 0.05.

**Figure 5 cancers-13-00687-f005:**
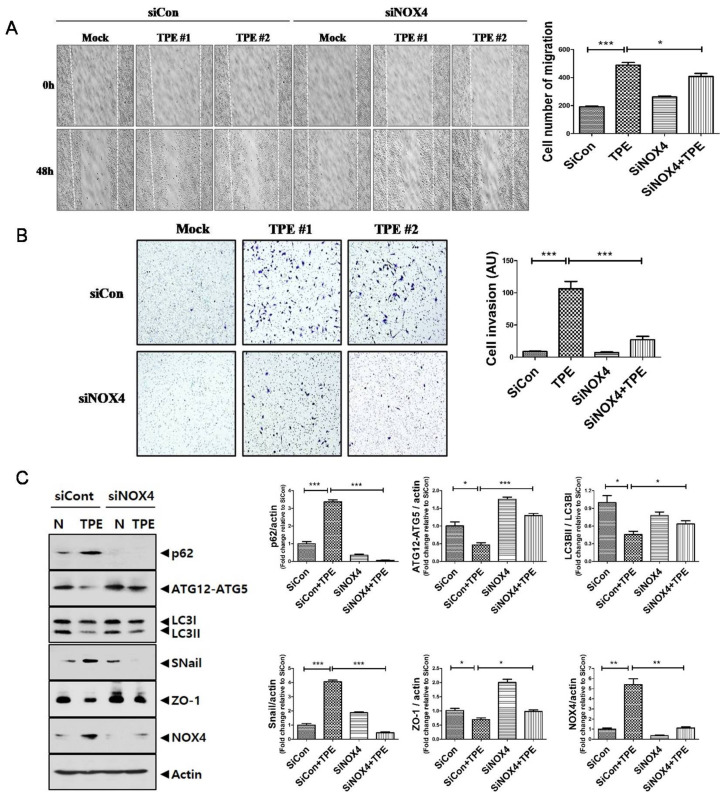
Tuberculous effusion enhances migration and invasion of lung cancer cell: Scratch-wound healing assay (**A**) and Transwell assay (**B**) showed thatNOX4 downregulation significantly prevented the migration and invasion of lung adenocarcinoma cells that were enhanced by TPE. (**C**) Western blotting of autophagy (P62, ATG5-ATG12, LC3II /LC3I) and EMT markers (Snail and ZO-1) in A549 cells with and without TPE. Data indicate the results of three experiments. *** *p* < 0.001, ** *p* < 0.01, * *p* < 0.05.

**Figure 6 cancers-13-00687-f006:**
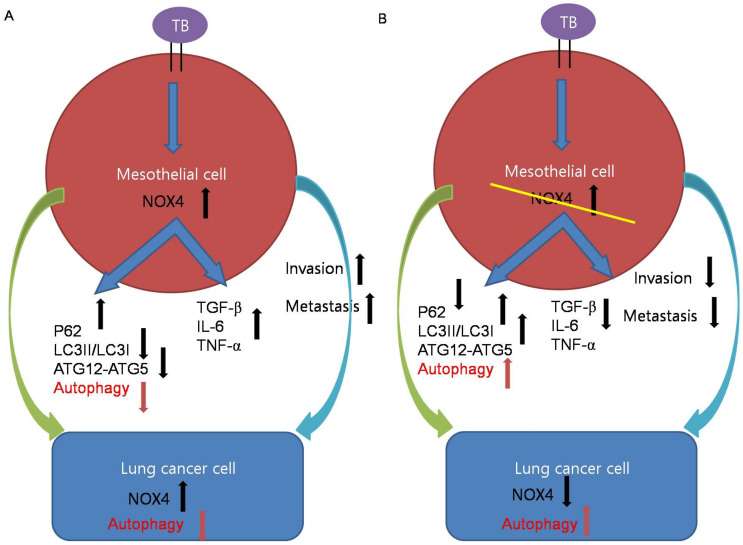
Schematic illustrations of the mechanism by which tuberculous fibrosis contributes to the malignant potential of lung cancer. (**A**) TB-infected PMCs exhibited increased expression of NOX4. NOX4 downregulates autophagy signaling and induces TGF-β, IL-6, and TNF-α, leading to cancer cell proliferation. In lung cancer cells after exposure to TB-infected PMCs, NOX4 signaling increases and autophagy signaling decreases. (**B**) Knockdown of NOX4 in TB-infected PMCs increases autophagy signaling and reduces TGF-β, IL-6, and TNF-α, leading to cancer suppression.

## Data Availability

The data used to support this research are available from the corresponding author upon request.

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
