# Peer review of "Tuberculous Fibrosis Enhances Tumorigenic Potential via the NOX4–Autophagy Axis"

_cancers, 2021, doi:10.3390/cancers13040687_

Round 1

Reviewer 1 Report

This is an interesting paper , somewhat difficult to read in the results, wich deserves to be published and needs minor typo corrections

Author Response

Thanks you for reviewing our manuscript.

Reviewer 2 Report

In the given study, authors investigated the role of NOX4 in tuberculous fibrosis-associated tumorigenicity both in vitro and in vivo. Mesothelial cells treated with heat-killed mycobacterium tuberculosis led to increased migration and invasive potential of lung cancer cells in a NOX4-dependent manner. Mice infected with BCG exhibited increased expression of NOX4 and enhanced malignant potential of lung cancer compared to mice injection with PBS while BCG+ KLN205 NOX4 KO mice group showed reduced metastatic potential of lung cancer, increased autophagy and decreased expression of inflammatory markers compared to the BCG+KLN205 WT mice group. Finally, NOX4 knockdown mitigated the malignant potential of A549 cells that was enhanced by tuberculous pleural effusion and restored autophagy signaling.Overall, these studies have identified a novel role of NOX4–autophagy axis regulated by tuberculous fibrosis  in enhanced tumorigenic potential and identified NOX4-P62 as a target for tuberculous fibrosis induced lung cancer.

This is an interesting study, is well designed, and most of the data supports conclusions made by the authors.

One minor comment that I have is:

Figure labels are small font so they are not legible, perhaps resolution and font size should be increased for clarity.

Author Response

As your suggestions, we increased the resolution and font size of figure labels.

Thank you.

Reviewer 3 Report

The manuscript by Woo et all describes the potential link between NOX4 and the increased incidence of lung cancer reported in patients infected with tuberculosis. The authors show that heat inactivated Mycobacterium tuberculosis results in an increase in NOX4 expression in mesothelial cells which was associated with an increase in autophagic markers and autophagic flux. Suppression of NOX4 ablated this phenomenon. Data is also presented that shows that tuberculosis infected cells promote the migration and invasion of lung tumor cells in a NOX4 dependent manner and lung cancer development is inhibited in the absence of NOX4 in a mouse model of tuberculosis. Human pleural effusions from tuberculosis patients exhibited similar effects on lung cancer cells. The manuscript in general is not well presented and is confusing in parts, and while this is a potentially interesting study and may support a role for NOX4 in the increased incidence of lung cancer in TB patients, the data presented is too preliminary to reach some of the conclusions made by the authors.

Specific comments:

  1. It is unclear why the authors choose to measure effects on EMT markers as well as autophagy markers, since the whole premise of the manuscript is that tuberculosis impacts lung cancer through NOX4 mediated increases in autophagy. The role of EMT in this process is not made clear in the manuscript and could be removed.
  2. In lines 92 and 93, the authors state that “These results suggest that NOX4 signaling mediates the EMT in tuberculous fibrosis by regulating autophagy signaling” in reference to the data presented in figure 1, but without showing effects on classical EMT markers, this statement is too premature.
  3. Similarly, in lines 203-206, the authors refer to TPE-induced EMT (with reference to the data presented in figure S2) yet no effects on classical EMT markers is presented.
  4. The authors should state upfront when discussing each figure, the relevance/rationale for measuring the effects on the markers in each figure, specifically POLDIP2 and collagen, TNF alpha, IL-6, TGF-beta, Snail, ZO1.
  5. The scratch wound healing assay presented in figure 2 is not too convincing and it is not easy to see the cells that have migrated into the wound.
  6. For the data presented in figure 3, the authors conclude that NOX 4 is required for increased metastatic potential of cancer cells by tuberculous fibrosis. However, tail vein injections of cells is not an accurate or true measure of metastatic potential. Cell injected this way naturally colonize the lung. In addition, the data presented in figure 3 shows that colonization of the lungs by tumor cells after BCG injection is reduced in NOX4 knockout mice compared to wild type mice. Do the wild type mice and NOX4 knockout mice develop BCG-induced pleurisy to the same extent? If not, the effects observed may solely be due to differences in BCG-induced pleurisy between the wild type and NOX. Some evidence of the pleurisy should be presented. Also, there is no mention of how many mice were used in each group. This needs to be stated.
  7. In the legend to figure 3, the authors state that BCG was administered subcutaneously, yet in the Methods section it states the BCG was administered in the intrapleural cavity. The authors need to be consistent and clarify which route of administration was used.
  8. It is unclear from the data presented in figures 4 and 5 how many patient pleural effusions were utilized. The authors state in the Methods that pleural effusions were collected from 5 patients, but do not make it clear in the figures how many were utilized. While it is understandable to only show representative images and data, it should be made apparent that the data obtained was similar from the majority of the patient samples used.
  9. Some sort of quantitation of figure 3D would help in making it more interpretable.
  10. Quantitation of the data in the western blot in figure 5C would help as some of the alterations in expression appear to be minor.
  11. The figures are far too small to be easily interpretable, especially the bar graphs, and these need to be enlarged.
  12. The figure legends are too vague and need to be more descriptive.

Author Response

Dear reviewer

Round 2

Reviewer 3 Report

The authors have satisfactorily addressed my concerns.